# Prescribed fire regimes influence responses of fungal and bacterial communities on new litter substrates in a brackish tidal marsh

**Viet Q. Dao** [1]ॐ*, **Crystal N. Johnson**[1]ॐ, **William J. Platt**[2]ॐ

1 Department of Environmental Sciences, Louisiana State University, Baton Rouge, LA, United States of America, 2 Department of Biological Sciences, Louisiana State University, Baton Rouge, LA, United States of America

ॐ These authors contributed equally to this work.
* daov94@gmail.com

**Data Availability Statement:** All code, spreadsheets, and DADA2 output ESVs are available in the Zenodo repository, doi: 10.5281/

## Abstract

Processes modifying newly deposited litter substrates should affect fine fuels in fire-managed tidal marsh ecosystems. Differences in chemical composition and dynamics of litter should arise from fire histories that generate pyrodiverse plant communities, tropical cyclones that deposit wrack as litter, tidal inundation that introduces and alters sediments and microbes, and interactions among these different processes. The resulting diversity and dynamics of available litter compounds should affect microbial (fungal and bacterial) communities and their roles in litter substrate dynamics and ecosystem responses over time. We experimentally examined effects of differences in litter types produced by different fire regimes and litter loads (simulating wrack deposition) on microbial community composition and changes over time. We established replicated plots at similar elevations within frequent tidal-inundation zones of a coastal brackish Louisiana marsh. Plots were located within blocks with different prescribed fire regimes. We deployed different measured loads of new sterilized litter collected from zones in which plots were established, then re-measured litter masses at subsequent collection times. We used DNA sequencing to characterize microbial communities, indicator families, and inferred ecosystem functions in litter subsamples. Differences in fire regimes had large, similar effects on fungal and bacterial indicator families and community compositions and were associated with alternate trajectories of community development over time. Both microbial and plant community compositional patterns were associated with fire regimes, but in dissimilar ways. Differences in litter loads introduced differences in sediment accumulation associated with tidal inundation that may have affected microbial communities. Our study further suggests that fire regimes and tropical cyclones, in the context of frequent tidal inundation, may interactively generate substrate heterogeneities and alter microbial community composition, potentially modifying fine fuels and hence subsequent fires. Understanding microbial community compositional and functional responses to fire regimes and tropical cyclones should be useful in management of coastal marsh ecosystems.

zenodo.13334170. https://zenodo.org/records/13334170.

**Funding:** VD & CN: NASA (National Aeronautics and Space Administration, USA: award number 80NSSC20M0216). https://www.nasa.gov/ CN: Louisiana Board of Regents (award number: LEQSF (2020-23)-Phase 3-14). https://www.laregents.edu/ The funders had no role in study design, data collection and analysis, decision to publish, or preparation of the manuscript.

**Competing interests:** The authors have declared that no competing interests exist.

## Introduction

Ecological processes affect the heterogeneity of fuels of both terrestrial and wetland ecosystems. In fire-frequented ecosystems, the fine fuels most likely to burn include standing or freshly fallen leaves and leaf litter, as well as other organic substrates that dry readily [1–3]. Amounts and composition of these fine fuels are broadly affected by the production and accumulation of organic plant tissues, as well as microbial decomposition of accumulated plant material in the resulting litter [4,5]. Tropical cyclones also may affect amounts and composition of litter substrates and hence fine fuels [6]. In addition, in wetland ecosystems such as tidal marshes, the fine fuels are periodically inundated and may be altered by litter and fine organic substrate deposition or removal [6]. These combined ecological processes should alter litter and fine fuels spatially and temporally, generating a diversity and wealth of compounds available for microbes [7]. In turn, the resulting substrate heterogeneity should modify how microbes influence litter substrate composition and dynamics. Resulting microbial-induced effects potentially could modify substrate dynamics and influence ecological processes, such as fire, in marsh ecosystems.

Fire and cyclone disturbances contribute to spatial variation in litter substrate heterogeneity in both terrestrial and wetland ecosystems. Low-intensity fires contribute to pyrodiversity that is reflected in mosaics of burnt, partially (and variably) burnt, and unburnt areas [8,9]. Pyrodiversity also reflects fire seasonality that affects plants and post-fire litter production [10] and is further influenced by recurrent fires that burn areas which may or may not have burnt in previous fires, resulting in local differences in numbers of fires within some period of time (i.e., patchy, unique fire frequencies) and altered litter consumption [11]. This pyrodiversity generates substrate heterogeneity from litter that is freshly burnt, converted to ash, charred, partially burnt, unburnt, and left in various stages of decay [12–14]. The litter heterogeneity is further diversified by tropical cyclones that move, mix, and deposit hurricane wrack and litter [15] across marshes and adjacent landscapes, increasing litter loads and influencing substrate nutrient interactions [16–18]. Together, resultant substrate heterogeneities should vary spatially, perhaps considerably, over areas affected by different disturbances.

Tidal patterns uniquely alter substrate heterogeneity in tidal marsh ecosystems. Tidal flooding modulates post-fire plant production of litter [19] and influences local temperature and oxic conditions [20] that may affect microbial transformations of the litter. Additionally, tides affect nutrient availability via adhesion interactions with suspended sediment [21], as well as through mixing effects [22]. Tides carry water and sediment [23] from distant locations [24] that could affect litter chemistry differently for newly deposited litter compared to existing, partially decomposed litter. The cumulative result should introduce further spatial and temporal variation in both substrate heterogeneity and microbial communities.

Ongoing changes in post-disturbance litter chemistry further develop/modify the substrate heterogeneity over time. The heterogeneity progresses through different stages characterized by unique litter qualities [25], C:N:P ratios [26], and recalcitrant compounds generated through decay and microbial decomposition [27]. These stages of post-disturbance heterogeneity affect plant community influence on litter chemistry [28] which may be additionally modified by the pyrodiversity generated from frequent fires [29]. Litter chemistry may also be affected by the mixing of newly deposited litter with existing litter that is progressing through various stages of decomposition. This mixing is further affected by the volume of newly deposited litter, which is rich in compounds missing from partially-decayed litter [30]. Thus, the resultant spatial heterogeneities should vary temporally over different stages of post-disturbance response.

Litter substrate heterogeneity affects community responses of resident microbes. Unique compounds in litter substrates may facilitate increases in abundance of both fungi and bacteria

with metabolic specializations for those compounds [31,32]. Microbial proliferation and microbial community composition are also affected by litter amounts [33]. Heterogeneity in the litter diversity can directly affect fungal community composition [34] by providing diverse compounds for fungal decomposition [35]. These include readily decomposed cellulose and more recalcitrant lignin, decomposition of which may involve different fungal communities [36]. Litter heterogeneity related to the balances of organic acids and cations may affect bacterial community composition [37]. Shifts in the availability of various compounds over time further develops litter heterogeneity and may continually affect community composition and decomposition functions of both fungi and bacteria [38–40]. Responses between fungal and bacterial communities may differ, as well as develop further along different paths as the heterogeneity shifts over time [41]. The litter heterogeneity ultimately resultant from interactions among fire and cyclone disturbances, tidal effects, and changes in these ecological processes over time should thus result in different fungal and bacterial community compositions.

Our study explored three hypotheses. These are centered on the concept that spatial and temporal variation in litter substrate heterogeneity in tidal marshes should result in dynamic local microbial communities, changing with environmental conditions and over time. We first hypothesized that differences in prescribed fire regimes (combinations of frequencies, times since most recent fire and seasonal timing of fire) should generate different microbial community compositions and also influence how those communities change over time. In particular, substantially different fire regimes should be associated with large differences in community composition. Second, differences in masses of newly deposited litter (hereafter, litter loads) should generate different microbial community compositions and also influence how those communities change over time. Third, fire regimes and litter loads should interactively generate different microbial community compositions and also influence how those communities change over time. The differences in community composition over time should relate to temporal variation in substrate heterogeneity associated with different times between sampling periods. Although these hypotheses focus on community compositional differences, there should also be differences in specific taxa and functions, any of which may extend to possible differences between fungi and bacteria that comprise the larger microbial community.

Our experimental design facilitated the study of effects of natural processes on microbial communities that develop on newly deposited litter and change over time. We examined both fungal and bacterial responses using plots in prescribed fire regimes that varied by fire frequency, and we characterized plant species in the plots in the different fire regimes. We placed sterile bags containing different amounts of new litter in plots within those fire regimes. Using samples collected at successive timepoints, we characterized dry-weight mass, fungal and bacterial community compositions, and inferred fungal and bacterial functions. We thus explored how fire regimes and litter loads, in conjunction with tidal inundation, influence substrate heterogeneity and the microbial communities that develop on new litter and change over time. We found significant differences in microbial community responses to fire regimes and subtle differences in them that further interacted with other ecological processes.

## Methods

### Study site

We conducted our study in the Big Branch Marsh National Wildlife Refuge (BBMNWR) in St. Tammany Parish, Louisiana, USA (30˚ 16' N, 89˚ 54' W). The refuge is located along the north shore and within the floodplain of Lake Pontchartrain, a shallow (average depth of ~3m), flat-bottomed fresh to brackish water lake, with salinity usually 1–10 ppt, but reaching 20 ppt during drought years [42]. The refuge is underlain by Pleistocene-aged consolidated

sediments, both marine sediments deposited during episodes of rising sea levels and fluvial sediments deposited during changes in Mississippi River direction [43]. St. Tammany Parish climate is warm-temperate, with 12°C average winter temperature, 5°C average daily minimum, periodic freezes, 27°C average summer temperature, and 33°C average daily summer high [44]. Total annual precipitation averages 155 cm, with 60% average relative daytime humidity [44]. During our study (mid-July 2022 to mid-December 2022), average daily temperature was 21°C, with 4°C minimum, 31°C maximum, and 54 days of recorded rainfall with 0.03cm minimum, 5.5cm maximum.

Along the north shore of Lake Pontchartrain, marsh habitats are zoned along a north-south <2m elevation gradient. Tidal marshes transition from salt marsh fringes along the lake edge to often-expansive, slightly brackish (oligohaline) marshes, to more inland fresh marshes, and finally to infrequently flooded, adjacent low-lying pine savannas in the most inland regions. These habitats are traversed by multiple north-south bayous and small tributaries [45]. Tides influence the marshes of Lake Pontchartrain; tides in the center of Lake Pontchartrain varied in height by ~0.9m on average during our study, with a minimum of -0.27m, and maximum of 0.66m [46]. All marshes in our study area, within BBMNWR, were inundated frequently by tides that often submerged litter substrates.

We conducted our study in the most inland oligohaline marshes of BBMNWR. These marshes are locally dominated by grasses (*Spartina patens* (Aiton) Muhlenberg), sedges (*Schoenoplectus americanus* (Persoon) Volk ex Schinz & R. Keller), and rushes (*Juncus roemerianus* Scheele). Other less abundant herbaceous species (e.g., *Typha domingensis* Persoon. *Sagittaria lancifolia* Linnaeus var. *media* Micheli, *Eleocharis cellulosa* Torrey, *Panicum virgatum* Linnaeus var. *virgatum*) typically occur in local patches. Nomenclature follows Rosen et al. [45].

Common disturbances of the BBMNWR include fires and tropical cyclones. The landscape has been managed using prescribed fires since 1994, with the primary goal to provide and maintain habitat for natural wildlife diversity, especially waterfowl and non-game migratory birds. Most marsh areas were burnt approximately every 2–3 years during the past several decades. Tropical cyclones crossed the region on average more than once a decade, and thus often affected substrate dynamics [47–49]. As noted in studies across the region, landfalling tropical cyclones tend to augment litter as wrack deposited in the marshes and adjacent pine savannas [50,51].

## Study design and plot establishment

Our experimental design was based on repeated measures of two treatments. These treatments reflected natural disturbances in tidal marshes: fire regime and tropical cyclones (increases in litter load that simulated wrack deposition). We established plots in oligohaline marshes in three different fire blocks in the marsh, each representing a different fire regime. Details of the study site, fire blocks, and plot locations are presented in [52]. Fire records informed fire regimes in fire blocks, considering fire frequency, seasonality, and time since fire. Fire regime treatments R1, R4, and R5 were burnt one, four, and five times, respectively, in the 10 years preceding the current study. The most recent fires prior to our study occurred 31 months (R1, December 2019), 15 months (R4, April 2021), and 20 months (R5, November 2020).

We established study plots within each fire regime in mid-July 2022 (day 0). All plots were 1x1m, selected to be randomly placed, independent, unpaired, and at 4m minimum distance away from other plots. Locations were chosen to place litter bags within 0.5m of the plot's center point, in visibly open sections of the litter layer, avoiding plant shoot clumps and bag overlap. Litter bags were anchored to the substrate at bases of plants with garden staples and

remained *in situ* until collection. Each plot received one of two litter loads, L1 (1x litter load) or L2 (2x litter load); four litter bags containing either 3.2 or 6.4g of litter were deployed at each plot on day 0. A total of 168 litter bags were deployed (four bags in 42 plots).

Plots were revisited after 60, 120, and 150 days to collect litter bags, designated D060, D120, and D150, respectively. Two bags were collected at D060, one bag at D120, and the final bag at D150. Bags were transported on ice and frozen at -20˚C. A minimum of 5 plots for each fire regime and litter load combination, with 3 or 4 bags retrieved from the majority of plots, totaled 133 total bags retrieved. S1 File presents the experimental, plot, and sampling design.

## Plant community and litter bag mass gain analyses

For each plot, the plant community was surveyed on day 0. Dominant plants were identified and abundance (cover) assessed in each plot. Generally, individual plots had one dominant species, but over the whole area within each fire regime there were one or two different dominant species. Additional species occurred at small abundances relatively rarely, with very little cover compared to dominant plants. Species were included in analysis if present in more than one plot, because rare species contributed little to litter composition. Thus, two plots were removed that were outliers with only one species not occurring in any other plot.

Because plant richness and composition were similar over most plots in a fire regime, we used a standard protocol for collecting freshly fallen litter for the field experiment. Standing dead plant material was manually extracted, avoiding partially decayed litter, in late June 2022. Litter was collected and pooled proportionally to the dominant plant community composition in each regime to avoid home-field advantage [53]. Thus, litter bag contents corresponded to the larger plant community and regime from which the litter was initially collected. Litter was collected away from plots to avoid interference with future litter bag deployment. Litter was transported to the lab and heated in an oven at 65˚C until dried. Litter was aseptically cut into pieces <3cm in length to approximate litter sizes in the marsh and size transformations produced by macrofauna excluded from bags. Litter was thoroughly mixed.

Litter bags were then prepared. The masses of nylon bags with 220μm mesh sizes with aluminum identifier tags were measured. Litter was added to reach 3.2g or 6.4g of litter. Bag masses were re-measured, bags sewn closed, and re-measured for final mass. Litter bags were shipped to the Radiation Science & Engineering Center at Pennsylvania State University and sterilized with gamma irradiation to a minimum dose of 39.4kGy. Once returned, litter bags were deployed in plots and later collected as described above. After DNA extraction, litter bag samples were heated in a 75˚C oven until dried and were weighed to quantify changes in mass. All material was kept in the litter bags to avoid loss of litter or sediment that contributed to mass gained. Final litter mass accounted for mass used in DNA extraction.

## Library construction and bioinformatics

Fungal and bacterial DNA was extracted and amplified. The DNeasy PowerSoil Pro kit (Qiagen, Germantown, MD) was used to extract DNA from 0.25g of litter. Litter was subsampled from multiple locations within each bag to generate a representative subsample of the larger litter bag sample. Extracted DNA was quantified on a Nanodrop 1000 (Thermo Fisher Scientific, Waltham, MA) and diluted to 1ng/μL for amplification. DNA was amplified according to instructions using the QIAseq 16S/ITS region panel (Qiagen) and amplified DNA from both ITS2 and 16S v4v5 regions of fungi and prokaryotes from the same 0.25g sample of litter. Amplification products were cleaned via magnetic bead purification using the same region panel. Using QIAseq index kits (Qiagen), sequencing barcodes were ligated to the amplified DNA with a second amplification stage and then cleaned.

Fungal and bacterial DNA was sequenced and processed into exact sequence variants (ESVs). Each sample containing indexed DNA product of both ITS and 16S regions was quantified, quality control assessed, and sequenced on an Illumina MiSeq 2000 by the LSU Health Sciences Genomics Core in Shreveport, Louisiana, USA, all to manufacturer protocols. One sample failed and was removed. There was a total of 132 samples each for ITS and 16S sequences, with water-only controls. The Qiagen CLC Workbench and Microbial Genomics Module were used to demultiplex and trim index barcodes, adapters, and primers from the Illumina reads. The Qiagen CLC Workbench output.fastq files separated by sample index barcode and by ITS and 16S sequences. Sequences were processed into ESVs using DADA2 v1.26 [54] according to the ITS v1.8 and 16S v1.16 pipeline tutorials, and classified using the UNITE v9.0 [55] and Silva v138.1 [56] databases. To control error, ESVs were discarded if they met any criteria: (1) the ESV was not identified to the kingdom level of Fungi, Archaea, or Bacteria, (2) over 10% of the total reads for that ESV was in negative controls, or (3) the ESV appeared in fewer than four of the 42 total plots.

## Statistical analyses

**Plant communities.** Plant species composition was compared across plots to evaluate differences associated with fire regime. Using plant relative abundance data, distances were calculated using Chord dissimilarity [57], ordinated using non-metric multidimensional scaling (NMDS) [58] with 100 tries and 2 dimensions, then analyzed via permutational multivariate analysis of variance (PERMANOVA, implemented with adonis2) [59] that tested the interaction of fire regime and litter load treatments. Homogeneous dispersion was checked and passed. The PERMANOVA analysis used Type I sums of squares with a significance level of $\alpha$ = 0.05. Additionally, dominant plant mean coverages with 95% confidence intervals were calculated; mean percentage coverages included all dominant species within each fire regime treatment.

**Litter mass.** Changes in litter bag mass were compared. Plots were removed from analysis if lacking measurements for every time point for that plot. One outlier plot was removed that had increases in mass >7x the average. General linear mixed modeling used Type III sums of squares and a significance level of $\alpha$ = 0.05 to compare differences in litter bag mass across the interaction of fire regime and litter load over time, and all main and 2-way interaction effects. Treatments were considered main fixed effects, with plot ID considered a random effect over which repeated measures were taken. Normality of residuals was tested by the Shapiro-Wilk test [60], and homogeneity of residual variance tested by the Breusch Pagan test [61]. Log transformations improved meeting assumptions and were applied, but did not fully resolve the assumption of normality of residuals. Average changes in mass with 95% confidence intervals and Tukey HSD comparisons were calculated over all plots in each combination of fire regime and litter load at each sampled time.

**Microbial communities.** Microbial community compositions were compared using ordination and PERMANOVA analyses. ESV data (abundances) within each sample (fungal and bacterial community considered separately) were transformed via the Hellinger distance transformation [62], and distances among communities were calculated based on Bray-Curtis dissimilarity [63]. Distances were then ordinated using another NMDS [58] with 100 tries and 4 dimensions. Using the same distance data, visualizations were generated of specific treatment combinations. Following ordination, a repeated measures PERMANOVA was conducted on the interaction of fire regime and litter load at different times and plant relative abundances to identify significant treatment effects on the fungal or bacterial community compositions. Plots were treated as strata to restrict permutations, thereby accounting for the variability associated

with repeated measures. The PERMANOVA model included additional nested random effect terms that account for the non-independent effect of time on plots, with plots nested within regime in agreement with the experimental design. This facilitated more accurate decomposition of residual variation and degrees of freedom into within-plot and among-plot variation. It also mitigated Type I errors resultant from inflated residual error degrees of freedom that affected other comparisons of model effects for significance [64,65]. Homogeneous dispersion was checked and passed. This analysis used Type I sums of squares and a significance level of α = 0.01 to account for the multiple analyses conducted on the microbial datasets.

Alpha diversity metrics were calculated and visualized separately but identically for both fungi and bacteria. Species richness [66], species evenness [67], and the Shannon diversity index [68] were calculated for each sample. Each was analyzed via generalized linear mixed models with Type III sums of squares and a significance level of α = 0.01. Species richness was modeled with a Poisson distribution and log link, species evenness with a beta distribution and logit link, and Shannon diversity with a gamma distribution and log link. Each linear model considered the interaction of fire regime and litter load at different times, with plot nested within fire regime as a random effect. Each metric's average with 95% confidence intervals were calculated over all plots in each combination of fire regime and litter load at different times.

Indicator species analysis was conducted separately but identically for fungi and bacteria. Indicator species analysis [69] uses differences in relative abundances of taxa associated with sites to identify taxa that can be used as markers of certain sites, community types, or changes in habitat or environmental conditions [70]. Family level was selected because many ESVs could not be identified to genus, but were identified to family, and because the family level provided more informative results compared to other levels. Indicator families were analyzed in the fire regime and litter load treatments, as well as in the combinations of fire regimes at different times. Analysis used group equalized point-biserial correlation coefficients. Indicator families were evaluated for significance using a significance level of α = 0.01. Further, to identify families with stronger association to a treatment combination, indicator families were considered for analysis if at least one correlation was greater than 0.35 in magnitude.

Functional compositions of the microbial communities were inferred and compared in each fire regime, litter load, and each fire regime at different sampling times. Using indicator families identified in association with the fire regimes, litter loads, and combinations between fire regime and time, categories of their ecosystem functions were inferred using FUNGuild [71] and PICRUSt2 [72] for fungi and bacteria, respectively. Only some ecosystem functions particularly related to biogeochemical processes were analyzed. PICRUSt2 used default settings and a minimum alignment of 0.8. PICRUSt2 outputs were interpreted and visualized using ggpicrust2 [73]. The ggpicrust2 analysis used the ALDEx2 differential abundance method and Benjamini-Hochberg correction. For each functional category, average relative abundances in each plot relative to all plots for that function, with 95% confidence intervals, were calculated. Calculations used read counts for fungal families and estimated functional abundances from ggpicrust2 for bacteria.

Contrasts were performed for significant effects identified in plant communities, litter bag mass gains, microbial communities, and alpha diversity metrics. These contrasts were based on orthogonal combinations of fire regime, litter loads, and time, and were R1 versus R4 & R5, R4 versus R5, L1 versus L2, D060 versus D120 & D150, D120 versus D150, and combinations of these contrasts. Contrasts used Type III sums of squares with a significance level of α = 0.01 when involving fire regime, litter load, or their interaction, or α = 0.0033 when involving effects over time, to account for the multiple analyses.

All statistical and mathematical analyses were conducted using Microsoft Excel and R version 4.2.2 [74]. The vegan package [75] was used for the Hellinger transformation, Chord and Bray-Curtis dissimilarities, NMDS, betadisper, adonis2, and alpha diversity metrics. glmmTMB was used for GLMMs [76], and DHARMa for model residuals [77]. Indicator species analysis was conducted using the indicspecies package [78]. Functional inferences used a Linux build of PICRUSt2 [72] and the FUNGuildR package in R [79]. This field research was authorized by written consent from the Big Branch Marsh National Wildlife Refuge, permit number 43558-21-11. The land is owned by the federal government and is protected. No protected species were sampled in this study. All code, spreadsheets, and DADA2 output ESVs are available in the Zenodo repository, doi: 10.5281/zenodo.13334170.

## Results

### Plant communities

The composition of plant communities differed among fire regimes. Plant community composition differed significantly among fire regime treatments (p = 0.001), but not between litter loads (p = 0.604) or by interactions among fire regimes and litter loads (p = 0.587) (S2 File). As indicated in Fig 1, there was clear and complete separation in the NMDS analysis (stress = 0.037) between fire regime treatment R1 and fire regimes R4 and R5. Many plots in R4 and R5 overlapped, reflecting similar species composition in the NMDS analysis.

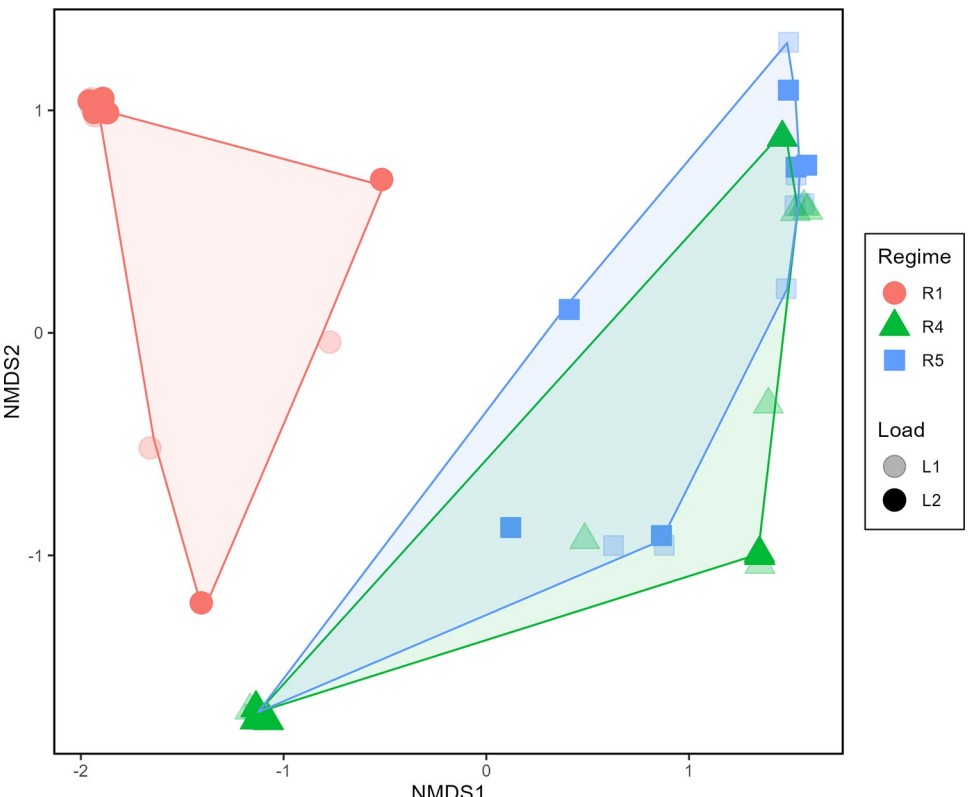

**Fig 1. Non-metric, multi-dimensional scaling (NMDS) ordination of plant community composition at the onset of the study.** Distances and ordination were based on plant relative abundance data derived from coverage. Plots (symbols) are coded by treatments: Fire regime (R) by symbol shape and color, and litter load (L) by symbol shading. Overlain polygons identify the spread and separation based on plant species composition of plots within the three fire regimes. Overlapping plots were slightly shifted from their actual location on the ordination to depict similar plots.

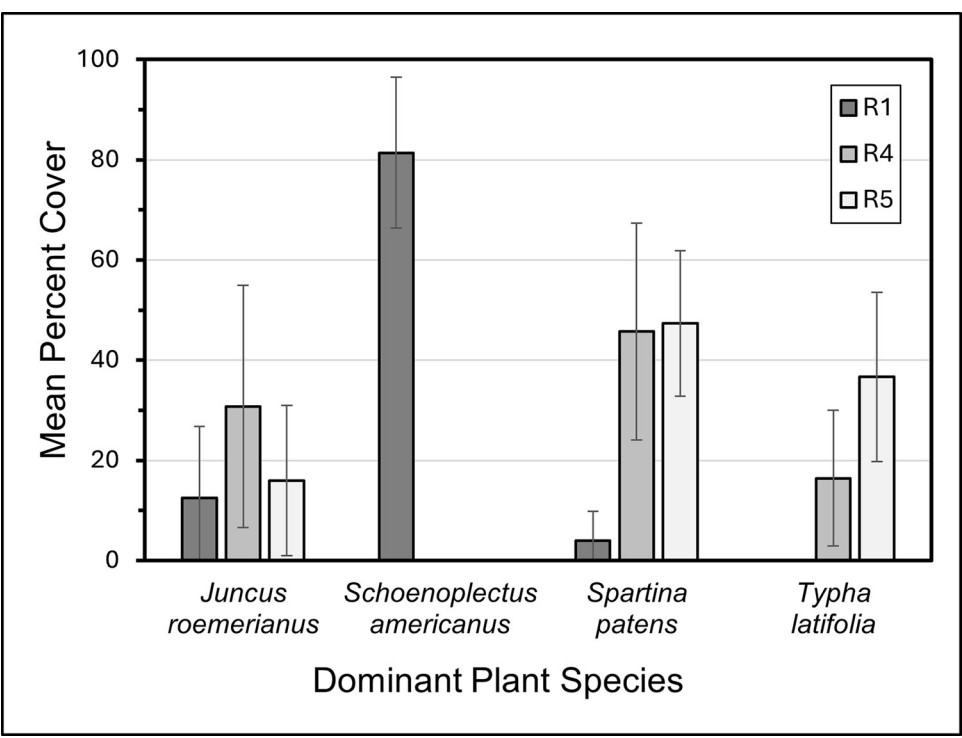

**Fig 2. Mean percent cover of the four dominant plant species within different fire regimes at the onset of the study.** Means (vertical bars) with +/- 95% confidence intervals (vertical lines) were calculated across all each species based on relative abundances in each plot within each fire regime treatment. Absence of values indicates the species was not present in any plots associated with a given fire regime.

Nonetheless, plant community composition differed in R1 from R4 & R5 (p = 0.001), but not between R4 and R5 (p = 0.218) (S3 File). The four most frequent plant species were *S. americanus*, *J. roemerianus*, *S. patens*, and *T. latifolia;* other species appeared at much smaller relative abundances. Dominant plant species varied across fire regimes (Fig 2). R1 was dominated by *S. americanus*, which did not occur in plots in R4 and R5; in contrast, *S. patens* was prominent in R4 and R5, but rarely occurred in R1.

## Litter mass

All litter bags increased in mass. Accumulated sediment was interlaced in the matrix of deployed litter, which also included some algal or plant root growth that contributed to mass gains. Masses increased by 0.9–5.3g (S4 File), with patterns of deposition differing by treatment. Mass gain differed by fire regime (p<0.001) and litter load (p<0.001). As illustrated in Fig 3, fire regime R1 had the highest overall mass gain at 3.20g +/- 0.16g (95% CI, confidence interval), which was significantly greater than mass gain for R4 and R5 at 2.56g +/- 0.12g, regardless of litter load (p<0.001). Mass gain was significantly greater (p<0.001) in the smaller (L1) than the larger (L2) litter load, with L1 gaining 0.76g +/- 0.13g more mass. Mass gain by litter bags was not different over time (p = 0.050), nor were there any significant differences associated with interactive effects of main treatments or interactive effects with time (S5 File). Results are detailed further in S4–S6 Files.

## Microbial communities

The dominant fungal phyla showed greatly different patterns across the three fire regimes. Of the 6 identified phyla, the most abundant fungal phylum was Ascomycota which was

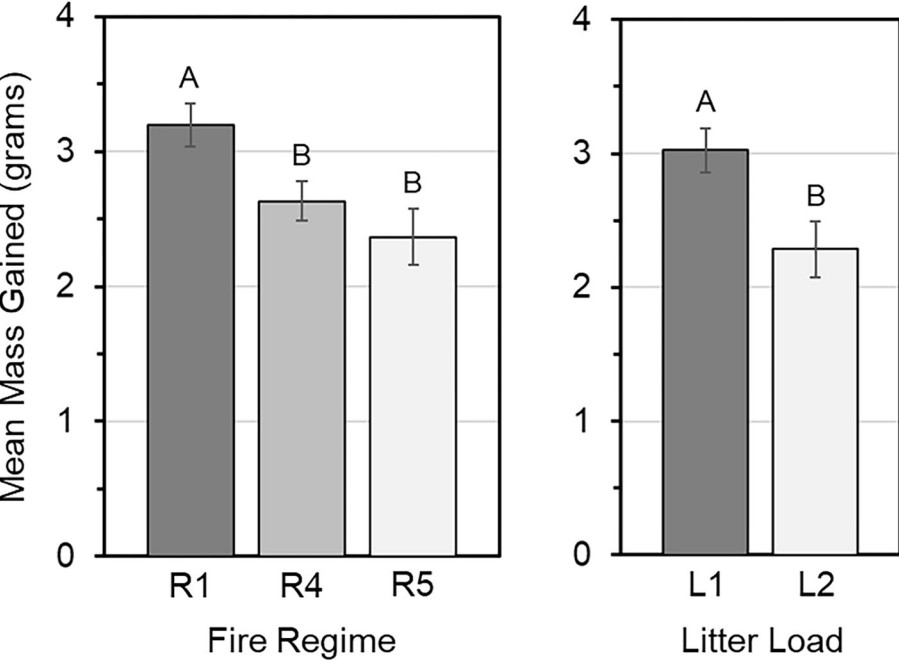

**Fig 3.** Average mass gain of litter bags in plots within different fire regimes (left) and with low or high litter loads (right). Means (vertical bars) and +/- 95% confidence intervals (vertical lines), with significant grouping letters from Tukey HSD pairwise comparisons, were calculated for all plots within each fire regime (R) and litter load (L). Within either fire regime or litter load, treatments with similar letters above bars are statistically similar in average percent mass gain.

overwhelmingly present in R1 and R4 (Fig 4A). By contrast, the relative abundance of Rozello-mycota was much greater in fire regime R5.

Patterns in bacterial phyla showed relatively consistent representation across the three fire regimes. Of the 13 identified phyla, Proteobacteria and Bacteroidota were the two most abundant phyla in all three fire regimes (Fig 4B). Qualitatively in R1, Acidobacteriota, Actinobacteriota, Fibrobacterota, Firmicutes, Spirochaetota, and Verrucomicrobiota were more represented. In R4, Bacteroidota were more represented, and Cyanobacteria were least represented. In R5, Chloroflexi and Planctomycetota were more represented.

Fungal community composition on the ESV level tended to be associated with fire regime treatments, but not clearly with litter loads. The fire regime effect was significant and there were different fire regime effects at different sampling times (p<0.001 for each) (S19 File). Fire regime, and the effect of different fire regimes at different times explained 21.99% and 4.16%, respectively, of community variation. Significant effects explained a total of 30.02% variation. Litter load was not a significant effect (p = 0.354) and explained little variation (0.64%). Major plants (*J. roemerianus*, *S. americanus*, *S. alterniflora*, and *T. latifolia*) together explained a total of 3.98% of fungal community variation. NMDS ordination (stress = 0.127) differentiated fungal communities clearly by fire regime, in which communities in R5 separated from those in R1 and R4 (Fig 5A). Visually, this separation is very different from the separation of plant communities in which R1 separated from R4 & R5 (Fig 1). When analyzed by time (Fig 5B), fungal communities in R1 and R4 exhibited minimal separation, but fungal communities in R5 tended to separate more distinctly with respect to time, with D060 separating most from D120 and D150 (cf. Panel A and B with Panel C in S7 File). Communities did not separate when

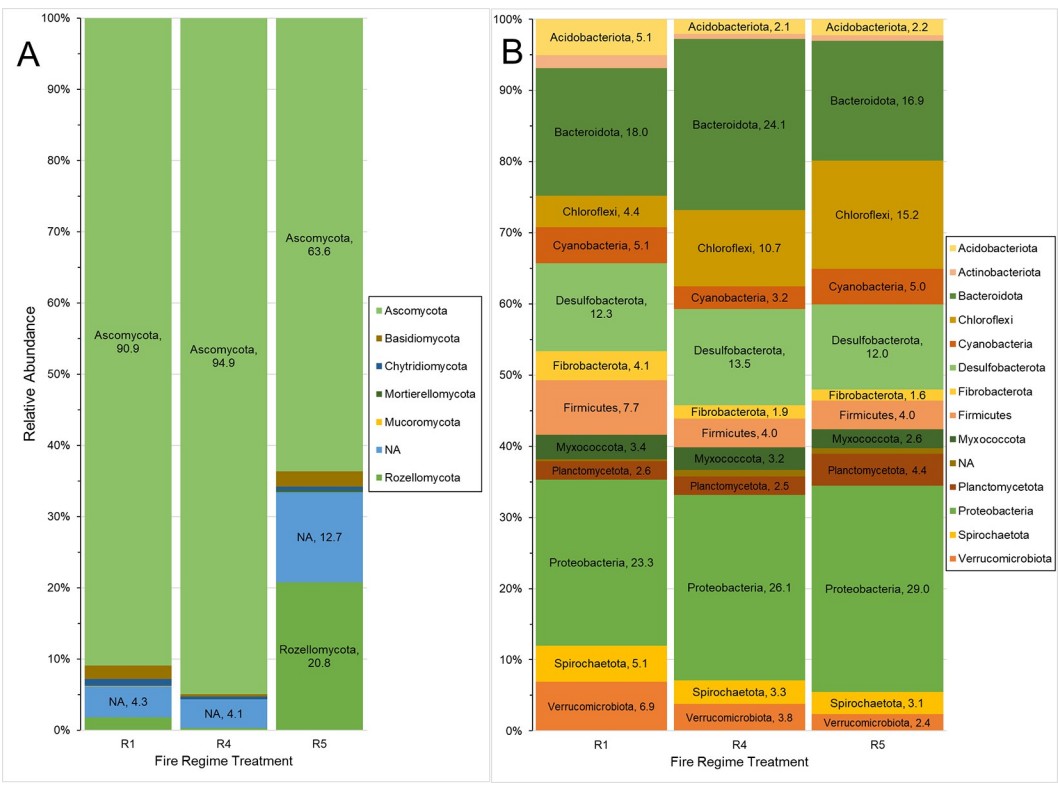

**Fig 4.** Percent relative abundances of fungal (A) and bacterial (B) phyla within different fire regimes. Relative abundances were calculated within individual fire regime treatments (R) so each bar represents the distribution of phyla within only that regime. Colors denote different phyla. The number next to a phylum indicates the abundance of that phylum relative to all phyla in that same treatment. NA denotes unidentifiable or unnamed phyla.

considering litter load (S8–S12 Files); however, communities in both litter treatments of R5 tended to be separated in time, with separation being greater in L1 than L2 (Panel E and F in S12 File). Contrasts supported NMDS ordination, in which fungal communities differed in R1 from R4 & R5, between R4 and R5, in R1 from R4 & R5 at time D060 compared to times D120 & D150, and between R4 and R5 at time D060 compared to times D120 & D150 ($p < 0.001$ for each).

Bacterial community composition also associated with fire regimes, but not clearly with litter loads. Fire regime, and the effect of different fire regimes at different times explained 26.14% and 4.57% of community variation, respectively (S19 File). Significant effects explained a total of 36.75% of variation. Litter load did not have a significant effect ($p = 0.2874$) and explained little variation (0.83%). *J. roemerianus* and *S. americanus* were significant effects ($p < 0.0013$ for each), and major plants explained a total of 4.28% of bacterial community variation. For the regime main effect, the NMDS ordination (stress = 0.058) differentiated bacterial communities in R5 from those in R1 and R4 (Fig 6A). This separation is also visually very different from the separation of plant communities in which R1 separated from R4 & R5 (Fig 1). Aside from some overlap with two plots in R1, bacterial communities in R1 separated from R4 (Fig 6A). When analyzed by time (Fig 6B), only R1 communities showed differences that transitioned from D060 to D120 to D150 (Panel A in S13 File). Communities did not strongly separate when considering litter load (S14–S18 Files), but there was some separation in some combinations of fire regime and litter load between D060 to D150 (Panel A, B, and E in S18 File). Contrasts supported NMDS patterns, in which bacterial communities differed in R1

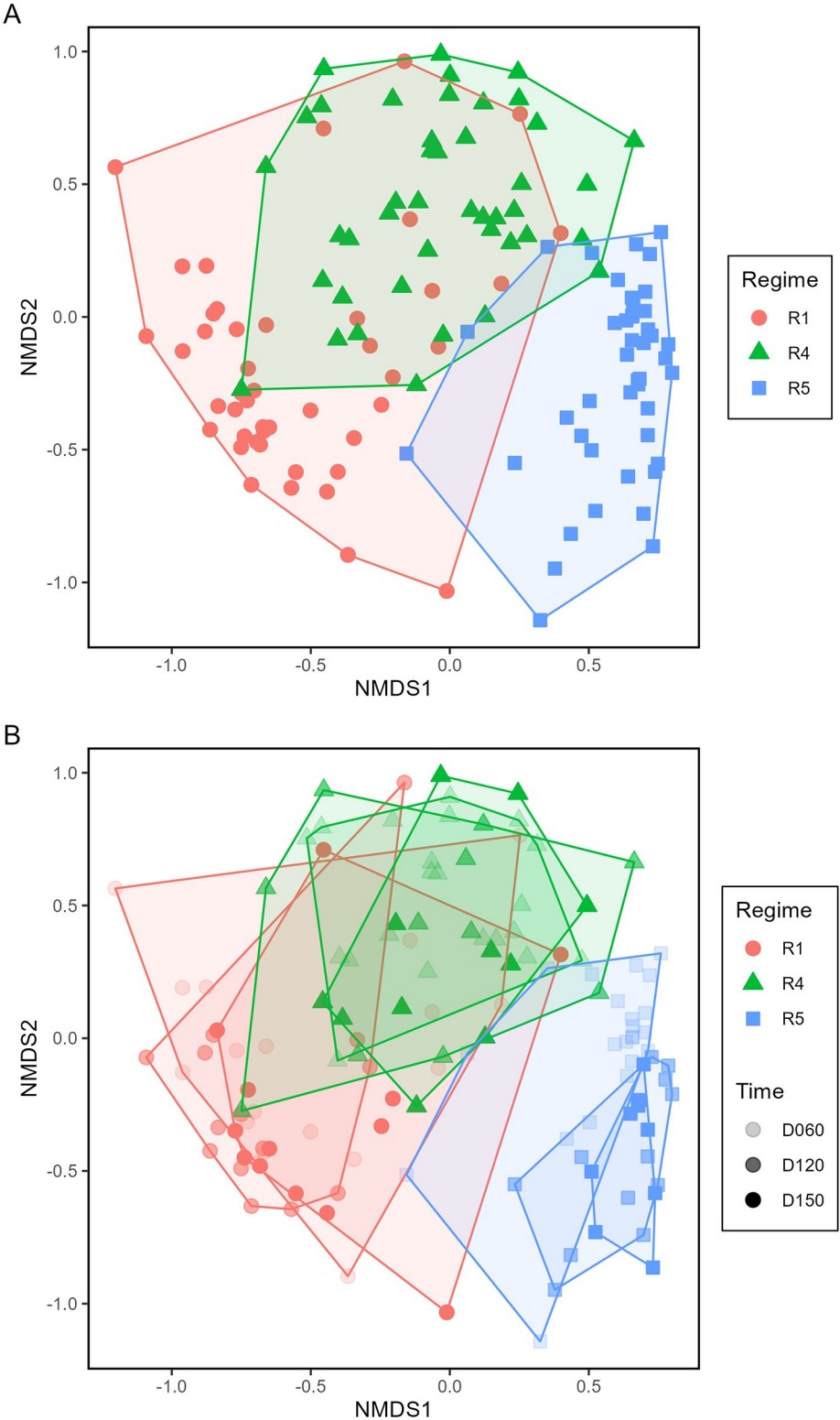

**Fig 5.** Non-metric, multi-dimensional scaling (NMDS) ordination of fungal community composition within different fire regimes (A) and within fire regimes at different sampling times (B). Distances and ordination were based on fungal ESV data in each treatment combination. Plots (symbols) are coded by treatments: Fire regime (R) by symbol color

and shape, and sampling time (D) by symbol shading. Overlain polygons identify the spread and separation of fungal species compositions in plots within the same treatment.

from R4 & R5, between R4 and R5, in R1 from R4 & R5 at time D060 compared to times D120 & D150, between R4 and R5 at time D060 compared to times D120 & D150 (p<0.001 for each), and finally between R4 and R5 at time D120 compared to time D150 (p = 0.0032).

Fungal community alpha diversity metrics showed patterns only in ESV richness. Metrics are presented in Fig 7 and S20–S22 Files. ESV richness differed by fire regime, litter load, the fire regime and litter load interaction, and by each effect with sampling time (p<0.0012 for each). Neither evenness nor Shannon diversity differed across treatments (p>0.054 and p>0.367, respectively). Richness specifically differed in R1 from R4 & R5, in R1 from R4 & R5 with different litter loads, between R4 and R5 with different litter loads (p<0.001 for each), and in some contrasts of sampling time in combination with fire regimes, litter loads, and their interaction (S3 File).

Bacterial community alpha diversity metrics showed different patterns among ESV richness, evenness, and Shannon diversity. Metrics are presented in Fig 8 and S20–S22 Files. ESV richness differed by fire regime, litter load, the fire regime and litter load interaction, and by each effect with sampling time (p<0.001 for each). Evenness differed by fire regime, litter load, and the fire regime and litter load interaction (p<0.0097 for each). Shannon diversity differed by fire regime and by fire regime with sampling time (p<0.001 for each). Richness specifically differed between R4 and R5, in R1 from R4 & R5 with different litter loads, between R4 and R5 with different litter loads (p<0.001 for each), and in some contrasts of sampling time in combination with fire regimes, litter loads, and their interaction (S3 File). Richness was generally higher in fire regime treatment R5 than in R1 or R4 (Fig 8A). Evenness specifically differed between R4 and R5, and in R1 from R4 & R5 with different litter loads (p<0.001 for each). Shannon diversity specifically differed between R4 and R5 (p<0.001), and in some contrasts of fire regime and sampling time.

### Indicator taxa

Many fungal and especially bacterial taxa were identified as indicators for particular fire regimes. Indicator families showed relative abundances associated with certain fire regimes and time points within a fire regime, but no families for either fungi or bacteria were identified as indicators of litter loads (S23 File). There were three and 81 fungal and bacterial families, respectively, identified as indicators for at least one fire regime.

Fungal indicator families with relatively strong associations to fire regimes were rare, and showed unique qualitative patterns in their relative abundances of inferred functions. Of the three fungal families that were indicators for at least one fire regime, two had moderate associations (correlations >0.35) with the infrequent fire regime R1, and one had a strong association (correlation >0.46) with the frequent fire regime R5 (S23 File). Regarding functions, the endophytic, pathogenic, and plant saprotrophic functions were consistently less represented in R4 or R5 than in R1 (Fig 9). Undefined saprotroph activity was particularly high in R5D060. Wood saprotroph activity was relatively consistent across fire regimes with no regimes showing particularly higher relative abundance.

Bacterial indicator families often associated strongly with a fire regime, and also showed unique qualitative patterns in inferred functions. Of the 81 bacterial families that were indicators for at least one fire regime, 27% had moderate and 17% had strong associations (correlations >0.35 or >0.5, respectively) with the infrequent fire regime R1 (S23 File). Additionally, 62% had moderate and 25% had strong associations (correlations >0.35 or >0.5, respectively)

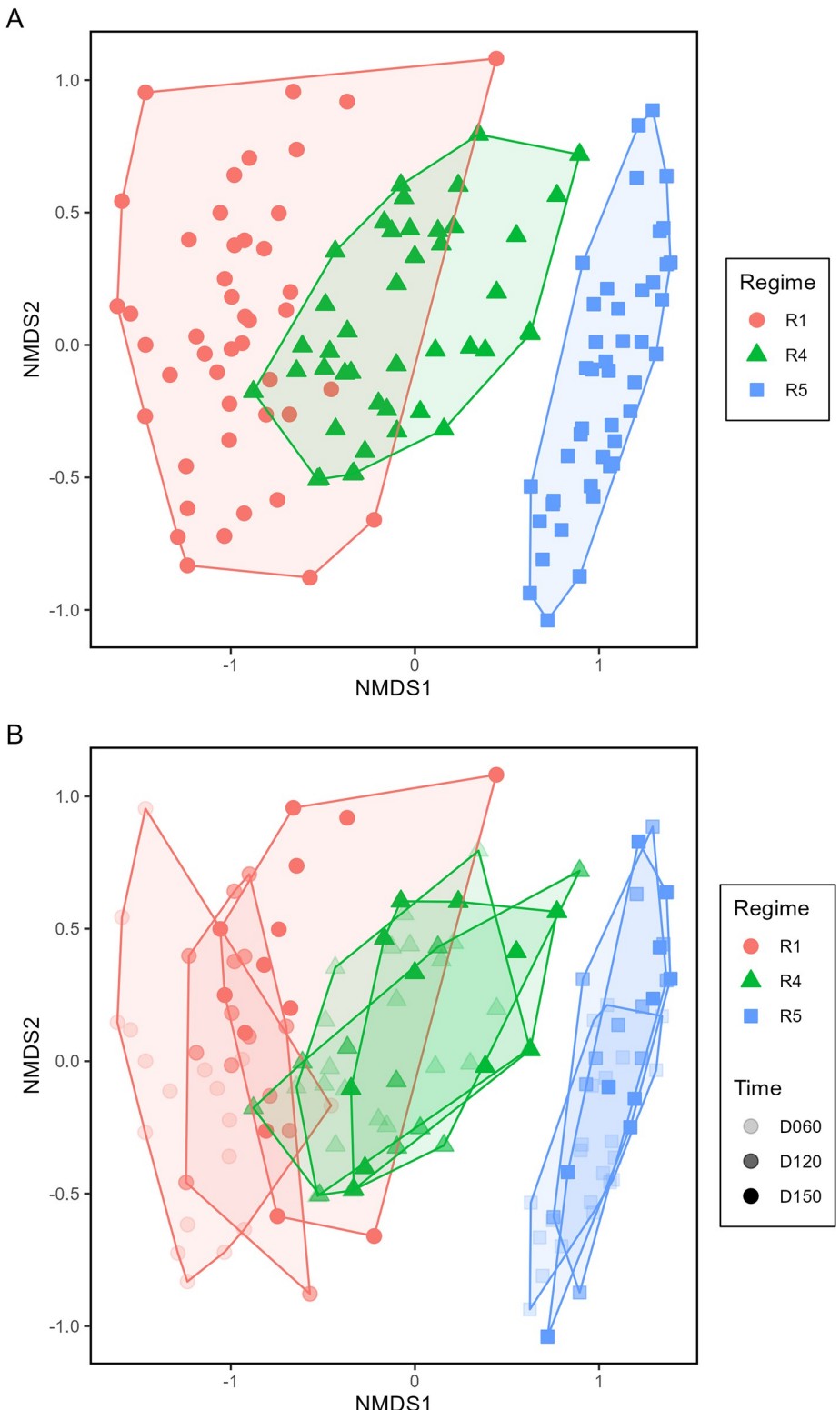

**Fig 6.** Non-metric, multi-dimensional scaling (NMDS) ordination of bacterial community composition within different fire regimes (A) and within fire regimes at different sampling times (B). Distances and ordination were based on bacterial ESV data in each treatment combination. Plots (symbols) are coded by treatments: Fire regime (R) by

symbol color and shape, and sampling time (D) by symbol shading. Overlain polygons identify the spread and separation of bacterial species compositions in plots within the same treatment.

with the frequent fire regime R5. The bacterial indicator families with the strongest associations (correlations >0.6) with R1 were Gallionellaceae, Koribacteraceae, and Sulfuricellaceae; meanwhile strongly associated with R5 were Azospirillaceae, Ancalomicrobiaceae, Hyphomicrobiaceae, Hyphomonadaceae, Hydrogenedensaceae, Pseudomonadaceae, Rhizobiaceae, and Sphingomonadaceae. Regarding functions, R5 interestingly had consistently higher average relative abundances of inferred functions than either R1 or especially R4 (Fig 10). Compared to other time points in a fire regime, relative abundances in D120 were highest for most categories in R5, whereas D150 was highest for all categories in R4. For the functions related to biogeochemical cycling, although there were differences by regime and time points within individual functions, the same pattern was generally replicated in fire regimes across functions.

## Discussion

Differences in microbial communities on newly deposited litter in tidal marshes of the Big Branch Marsh National Wildlife Refuge were strongly associated with fire regimes. The composition of both fungal and bacterial communities, as well as changes in those communities over time, varied significantly among fire regime treatments. Further, fungal richness, bacterial richness, bacterial evenness, and bacterial Shannon diversity differed by fire regimes, and all of these except bacterial evenness differed among fire regimes over time. In contrast, differences in litter load or in litter loads over time were not directly associated with differences in microbial communities. Nonetheless, interactive effects of fire regime and litter load on microbial community composition, in the context of frequent tidal inundation, were weakly supported by the data from our study. These results, strikingly similar for fungal and bacterial communities, indicated that litter dynamics should be influenced by differences in fire regimes and resulting responses of pyrophilous microbes in the subtle context of differences in litter loads in environments modified by tropical cyclones that occur in the context of frequent tidal inundation. Detailed results relative to each of our hypotheses are summarized in the S24 File.

### The consequences of differences in fire regimes

Diverse fire regimes were associated with differences in composition of both fungal and bacterial communities. Differences in fire regimes may change the establishment of microbial communities on newly deposited litter substrates [80], resulting in divergent trajectories of community development [81]. We anticipate that indicator families strongly associated with different fire regimes may be the main drivers of divergence in composition. For example, of the three fire regimes we examined, two fire regimes, R4 and R5, had similar fire frequencies of 4–5 fires in 10 years preceding our study. This contrasted with fire regime R1 that was burnt once in 10 years, and also had the longest time since fire compared to all fire regimes. These large differences, especially in fire return intervals, could have generated substantially different heterogeneities that may explain why microbial community compositions in R1 were different from R4 & R5 at the different sampling dates. These community differences associated with substantially different fire frequencies support the findings of others [82].

Subtle differences in fire regimes may associate with substantive differences in microbial community composition, especially for bacteria. Fire regime R5 experienced the shortest fire return intervals in the period preceding our study, as well as the second most recent fire. By

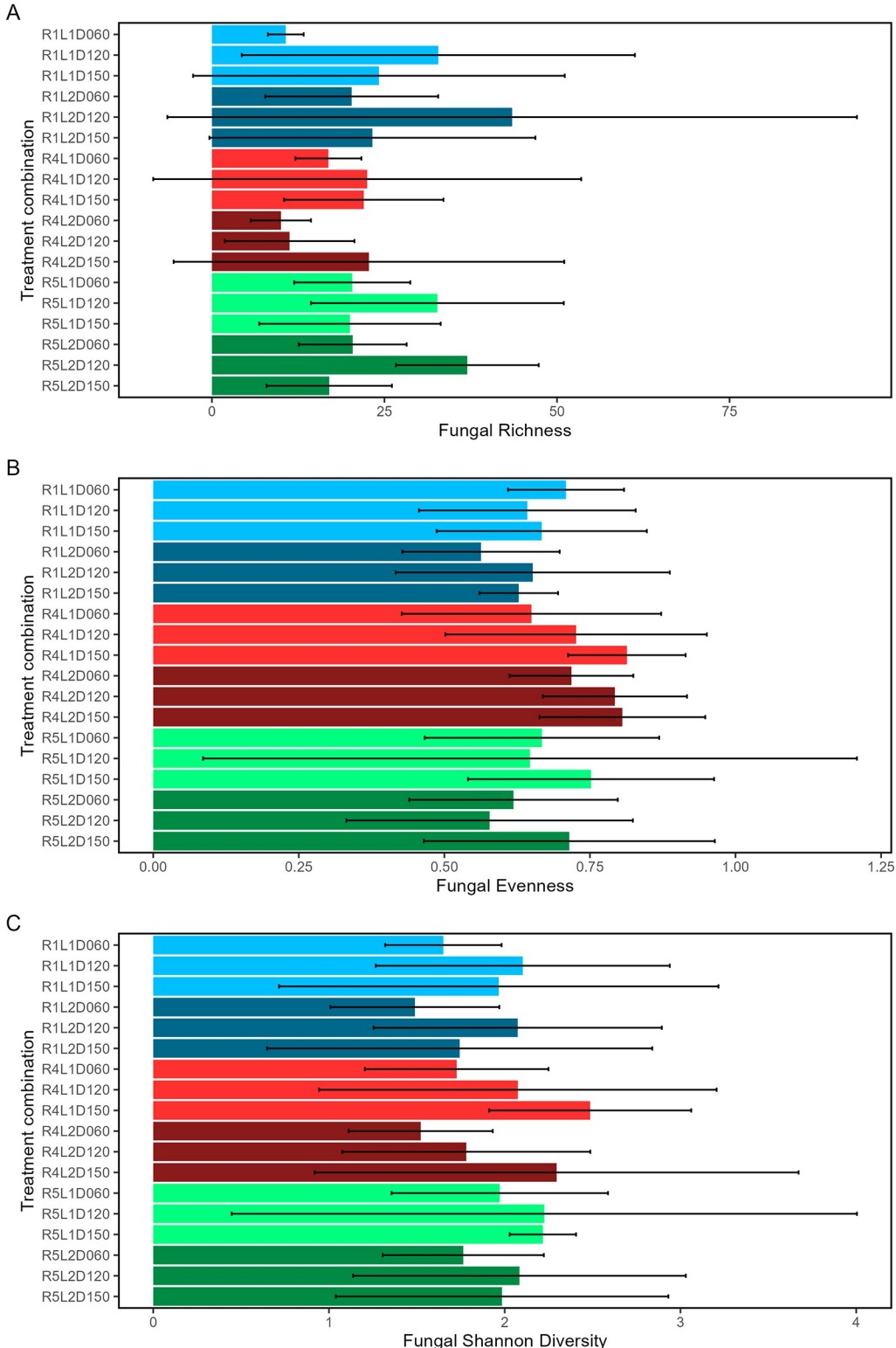

**Fig 7.** Fungal alpha diversity metrics of species richness (A), evenness (B), and Shannon diversity (C) within combinations of fire regimes and litter loads at different sampling times. Means (horizontal bars) and +/- 95% confidence intervals (horizontal lines) were calculated across all plots in each treatment combination of fire regime (R), litter load (L), and sampling time (D). Fire regime and litter load are coded by color.

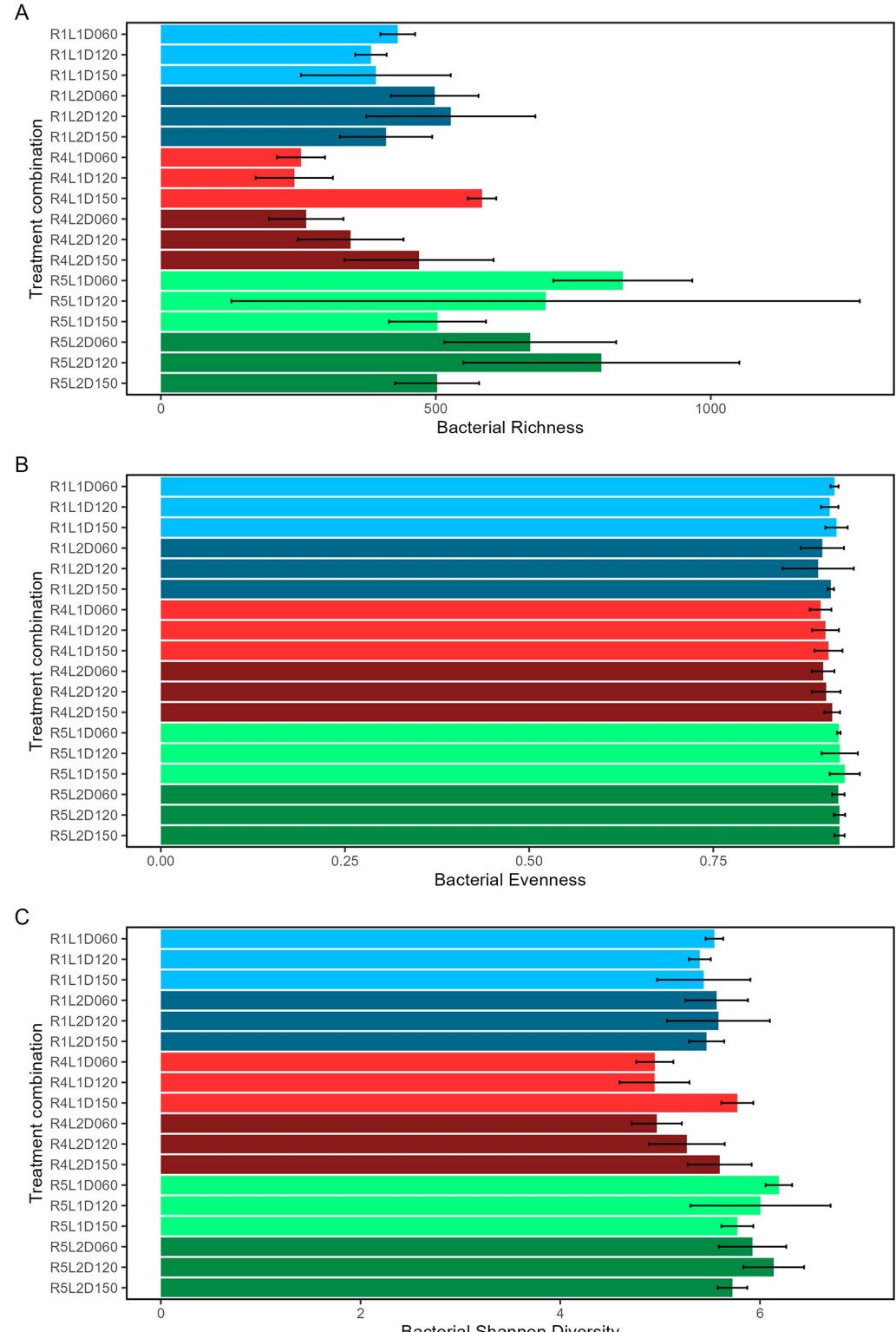

**Fig 8.** Bacterial alpha diversity metrics of species richness (A), evenness (B), and Shannon diversity (C) within combinations of fire regimes and litter loads at different sampling times. Means (horizontal bars) and +/- 95% confidence intervals (horizontal lines) were calculated across all plots in each treatment combination of fire regime (R), litter load (L), and sampling time (D). Fire regime and litter load are coded by color.

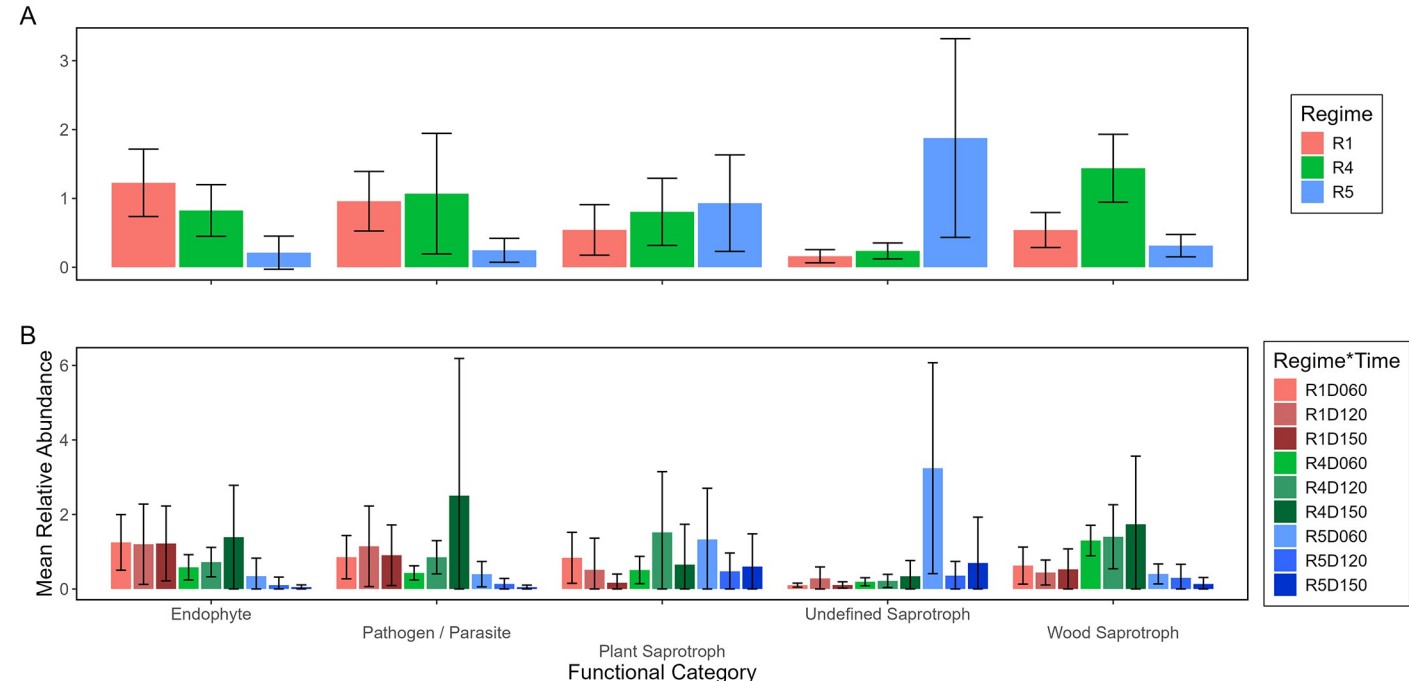

**Fig 9.** Average percent relative abundances of inferred functions of fungal indicator families within different fire regimes (A) and within fire regimes at different sampling times (B). Mean (vertical bars) and +/- 95% confidence intervals (vertical lines) were calculated as the average of all plots' relative abundances of a particular function within a particular fire regime (R) or fire regime at a particular sampling time (D). An individual plot's relative abundance of a particular function was calculated relative to all plots in the system for that particular function. If a confidence interval's lower bound was negative, it was truncated to -0.1 to maintain plot clarity. Fire regime and sampling time are coded by color.

comparison, fire regime R4 experienced slightly longer fire return intervals in the period preceding our study, as well as the most recent fire. Despite the similarity in these two fire regimes, there were particularly large differences in microbial community compositions, with nearly, and fully complete separations that occurred between R4 and R5 for fungi and bacteria (Figs 5 and 6, respectively, as well as individual degrees of freedom contrasts). Bacterial ESV richness strongly differed between R4 and R5, with the majority of bacterial indicator families strongly associated with either R4 or R5. These results suggest that bacterial communities in particular may show potentially strong species and relative abundance differences between generally similar fire regimes. We suggest that even similar fire regimes may generate enough spatial and temporal variability in substrate heterogeneity to influence community differences in composition.

Differential changes in fire regime effects over time (the initial 150 days after deposition of new litter) likely influenced patterns of microbial community responses. Fungal and bacterial community compositions were associated with fire regime effects that varied by sampling time points. Further, microbial communities were more separated at D060 than at D120 & D150, compared to the separation between D120 and D150. These patterns suggest that the fire regime effects may be modified by temporal variation over the 2 months of potential development between D060 and D120 that was more than the variation over the 1 month between D120 and D150. Many indicator families were strongly associated with particular time points in a fire regime, and they may have responded to the heterogeneity at that specific time. The effects of those fire regimes likely shifted over time. Temporal changes in these indicator family associations likely drove community composition changes. These findings are consistent with

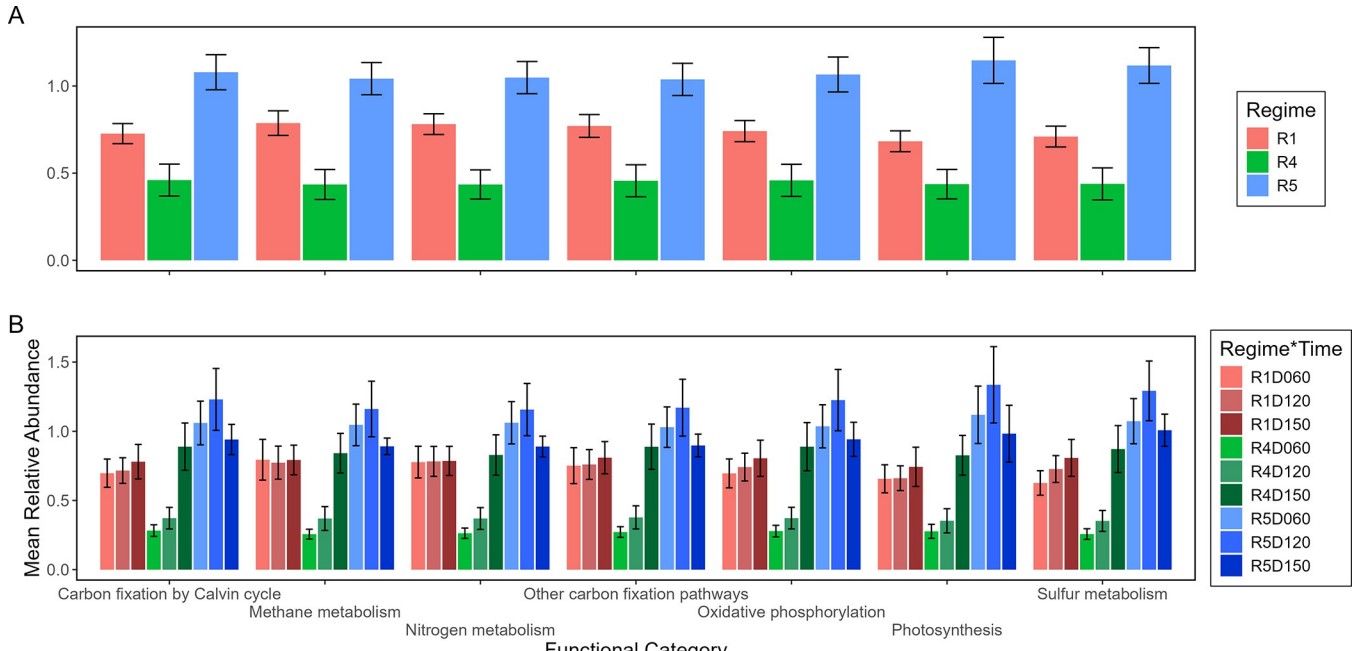

**Fig 10.** Average percent relative abundances of inferred functions of bacterial indicator families within different fire regimes (A) and within fire regimes at different sampling times (B). Mean (vertical bars) and +/- 95% confidence intervals (vertical lines) were calculated as the average of all plots' relative abundances of a particular function within a particular fire regime (R) or fire regime at a particular sampling time (D). An individual plot's relative abundance of a particular function was calculated relative to all plots in the system for that particular function. Fire regime and sampling time are coded by color.

other findings that microbial communities differed with time since fire [83,84]. Thus, ongoing changes in effects of diverse fire regimes over time potentially influence substrate heterogeneity and result in different microbial communities.

Microbial communities showed responses to the fire regime effects that were somewhat disconnected from direct plant community differences. Both plant community and microbial community compositions differed between R1 and R4 & R5, suggesting strong effects of fire regimes, especially fire return intervals, on community compositions of ground layer organisms, including plants, fungi, and bacteria. Nonetheless, there were divergent patterns in composition of plant and microbial communities between fire regimes R4 and R5, as shown in the NMDS ordinations and contrasts. Although plant communities were similar between R4 and R5, microbial communities strongly differed between these two regimes, with bacteria in particular showing complete separation on the NMDS (Fig 6). Additionally, little variation in microbial community composition was explained by plants, suggesting that, although *J. roemerianus* and *S. americanus* had significant effects on bacterial community compositional differences, plants had relatively little influence on composition of microbial communities as a whole.

Microbial communities may respond to heterogeneity produced by fire regimes that interact with indirect plant effects. Heterogeneity may be influenced by indirect plant effects through litter composition [12,39], leaf nitrogen content, cycling, leaf cover, and leaf production [85], lignin content and C:N ratio [86], litter chemistry [87,88], as well as fire-induced changes in plants and plant litter characteristics [86,88]. These indirect plant effects on heterogeneity may influence a microbial taxon differently than more direct plant community composition effects. This may be the case for Didymosphaeriaceae which has endophytic and saprotrophic roles [89] and was identified as an indicator family in R1 and R5 where

endophytic and saprotrophic functions were each relatively abundant. Although direct plant effects are possible with their endophytic functions, the lack of overall plant community effects suggests this family also responded with its decomposition functions to heterogeneity that may be associated with indirect plant effects.

### Litter substrate dynamics in tidal marshes

Tidal marshes generated nuanced effects associated with sediment accumulation. Changes in litter substrates after deposition appeared to be influenced by differences in fire regimes, which were associated not only with different plant communities, but also differences in litter composition. These different litter compositions had varying physical traits (surface area, adhesion, interior space and arrangement) that potentially affect the accumulation of sediment introduced via tidal patterns. Such effects might explain why R1 had the higher mass gains than R4 or R5. The accumulation of sediment was likely further affected by different litter loads that result in different available volumes to accumulate sediment. In this way, our study suggests that tropical cyclone effects on litter loads may further modulate the interactions involving sediment accumulation already influenced by fire regimes, plants, litter physical traits, and tidal patterns. The net effect of accumulated sediments may constitute a subtle spatial influence on substrate heterogeneity and hence on microbial community composition in tidal marshes.

Some microbial community responses may be explained by the sediment accumulation that varied with interactive effects of fire regime and litter load. For fungi in R1 and R4, communities in one litter load somewhat clustered within communities of the other litter load (cf. Panel A and B in S11 File). Additionally for bacterial communities in R5, most plots clustered separately between litter loads (cf. the locations of the main groups of plots between litter loads in Panel C in S17 File). These patterns agree with others who found that tidal-suspended sediments drove divergence in microbial communities [90]. The interaction of litter load with sediment accumulation and fire regimes, in regions where tropical cyclones are prevalent, may have indirect but pervasive effects on microbial communities.

Sediments may facilitate microbial introduction via tidal patterns that directly influence microbial communities. The initial introduction could be a key mechanism of tidal influence on newly deposited litter. These initial microbes likely form the first group of species that merge with the communities associated with plant-attached leaves and strongly affect early stage community composition [91,92]. Additionally, tides continuously introduce other microbes that compete with previously established microbes [92] to influence the trajectories of community composition.

Indirect tidal effects may continually influence the substrate heterogeneity and microbial communities. Tidal patterns and sediments alter local salinity [93], litter nutrient availability [94], and oxygen and redox potentials [95] that further develop over time with repeated inundation. The altered environment may then affect microbial community composition and activity [96]. These different communities can potentially produce vastly divergent heterogeneities over time [97] and in turn influence community compositions. This may partially explain the large community differences between R1 and R4 & R5 since R1 had different fire effects and more sediment accumulation. These findings support others who found that microbial communities were influenced by temporal variation in fire- and tide-altered heterogeneity [98,99]. The effects of litter loads from tropical cyclones thus likely affect substrate heterogeneity and microbial communities in tandem with the effects of tidal patterns and the litter compositions resultant of fire regime and plants.

## Pyrophilic microbes

Pyrophilic microbes may respond to heterogeneity associated with high frequency fire regimes. In our study, multiple indicator taxa had strong associations with the higher fire frequency fire regimes. Other studies also have identified pyrophilous indicator taxa associated with post-fire effects [100–102]. For example, the fungal family Pleosporaceae was associated with the frequently burnt fire regimes R4 and R5, which supports other findings [4]. Their saprotrophic lifestyle [103] may facilitate responses to post-fire substrate heterogeneity in frequent fire regimes. Additionally, bacterial family Oxalobacteraceae was strongly associated with R5, supporting other findings of their association with high frequency fires [104] and higher frequency fire regimes in the litter [7]. The R5 substrate heterogeneity may be preferential for their carbon and cellulose decomposition that encourage post-fire plant growth [105].

Pyrophilic microbes may influence community composition and post-fire response over time. Some pyrophilic families may have post-fire adaptations and growth strategies, including diverse resource usage [106], faster growth rates [107], and alternative competition dynamics at different stages [108] that facilitate associations with particular stages of post-fire response. The newly deposited litter may host post-fire early-stage substrate heterogeneity to which certain pyrophilous microbes may be adapted [101] and establish following their introduction from tides or buried litter [109]. Certain pyrophilous microbes may remain established and continue influencing larger community composition over time [110]. This may be the case for bacterial family Azospirillaceae that strongly associated with R5 and had the highest correlations at D060 but also moderate correlations at D120 and D150. These results support others who found that members of Azospirillaceae were enriched by burning and remained at higher levels compared to unburnt controls throughout sampling periods after fire [111]. Additionally, the bacterial family Hyphomicrobiaceae showed strongest associations to only earlier time points in R5, which supports others who found their enrichment in a burnt forest ecosystem [112] and soils enriched by biochar [113]. Both Azospirillaceae and Hyphomicrobiaceae appear to be responsive to post-fire environments in frequently burnt systems, where their roles in nitrogen cycling may influence larger post-fire ecosystem response. Differences in fire regimes and resultant heterogeneities thus appear to facilitate different responses by pyrophilic taxa, which likely affect ecosystem functions.

## Ecosystem functions

Even subtle differences in fire regimes may affect fungal functions. Fungal saprotrophy and decomposition of litter appear responsive to frequent fires in terrestrial systems [29,39,88], and were clearly different in tidal marsh environments as shown by the varying predominance of saprotrophy guilds depending on fire regimes applied. Fungal communities and functions were also different between frequently burnt fire regimes R4 and R5, suggesting sensitivity of potentially pyrophilic fungi to even subtle environmental differences associated with fire regimes. This sensitivity to fire regime likely explains why plant, undefined, and wood saprotroph patterns were not consistent across the three fire regimes, and likely relate to specific fungal taxa that perform each function.

Bacteria also may have functional redundancy and resilience sensitive to subtle differences in fire regimes. Even though there were community compositional differences among fire regimes, many bacteria in the different fire regimes still perform similar functions. This functional redundancy may explain why the relative abundance of one function in a fire regime was consistent with another function in the same fire regime, and supports other findings that bacterial functional genes in carbon and nitrogen cycling were present across different fire affected sites [114]. The redundancy may be important for this marsh ecosystem that

experiences frequent tidal flooding because although different bacteria change in abundance differently with respect to flooded conditions, overall they still perform substantial amounts of nutrient cycling and decomposition [115]. Separate from redundancy, the stark difference in functional abundance between the frequently burnt fire regimes R4 and R5 suggests that subtle differences in fire regimes may still greatly affect communities and their functions. Although the functions may be present in both fire regimes, the substrate heterogeneity in each fire regime may not facilitate consistent growth of the bacteria or consistent abundance of their functions.

The responses of pyrophilic indicator families to fire regimes may have important effects on post-fire ecosystem response. Pyrophilic indicator families may facilitate important ecosystem functional redundancy and resilience in frequently burnt ecosystems subject to high frequency fire regimes [116]. Their increased nutrient cycling capabilities can influence post-fire regrowth of the plant community [117], which may be particularly important with respect to off-fire-season fires that can have complex, variable effects on plant communities [10,118]. However, the responses of pyrophilic taxa show sensitivity to even subtly different fire regimes, and highlight the importance of maintaining fire regimes for natural ecosystem processes.

## Litter loads and future fires

Post-fire responses of pyrophilic microbes may have ramifications on future ecosystem management by affecting litter fuel loads and future fires. This could influence fire regimes established in pursuit of management goals and have potential impacts on plant community structure [119]. For example, certain changes in pyrophilic fungal growth in response to different fire regimes may increase litter decomposition and negatively affect the likelihood of future fire. Separately, changes in pyrophilic plant growth promoting microbes may affect nutrient cycling, plant growth, and litter production. These microbes assist plant growth by providing nutrients [120] or by directly recycling litter nutrients back to the plant [121]. Because upwards of 75% of all plant nitrogen and phosphorus come from microbes, alterations in that nutrient flow for plants may bring profound influence on plant growth, productivity, and litter production [122]. Increases in pyrophilic indicator families may result in increased nutrient cycling and decomposition that may increase fuel loads and positively affect likelihoods of future fires.

## Conclusions

Microbial communities that developed on newly deposited litter in our studied tidal marsh clearly responded to differences in fire regimes. Composition of fungal and bacterial communities also was influenced by indirect effects due to variation in plant communities, sediments, and changes in those effects over time. Microbial community responses were somewhat decoupled from plant community responses to fire regimes, showing unique patterns in response to similar fire regimes at short fire return intervals. Our findings identify the nuanced influence of even small variation in fire regimes on litter substrate heterogeneity and differences in microbial communities associated with each regime. As a result, generalizations should be made carefully, as even nuanced differences can be associated with large changes in microbial communities.

Spatial and temporal variations in substrate heterogeneity also are likely to affect microbial community responses. First, fire regimes, cyclone effects through litter loads, and tidal patterns that introduce sediment effects, combine together to generate spatial variation in substrate heterogeneity. Second, ongoing modifications to the litter environment from microbial decomposition and nutrient transformations, cyclonal mixing, and tidal sediment effects, likely are

combined to generate temporal variation in substrate heterogeneity. Tides may further directly influence community compositions by facilitating the introduction of microbes to new litter substrate, and by modifying the existing substrate heterogeneity. Certain microbes, especially pyrophilic indicator taxa, likely have adaptations that facilitate better growth in particular heterogeneities to affect larger community response in the different fire regimes.

We identify an ecosystem-level relationship focused on the dynamics of litter produced by plants that are acted on by the microbial communities and their roles in decomposition and nutrient cycling. In tidal marshes, those dynamics are subjected to the effects of frequent fire, as well as by tidal influences and periodic effects of tropical cyclones. Litter dynamics thus become influenced by a panoply of additional ecological processes that generate heterogeneity in the litter and its spatio-temporal dynamics. In our study, we show that this heterogeneity is associated with a diversity of fungal and bacterial communities, and these communities are responsive to even subtle changes in fire regimes.

Understanding how microbial communities and their functions respond to fire regimes should facilitate achieving certain goals of adaptive management. Future research is needed to quantify the influence of subtle changes in ecosystem processes and disturbances on microbial functions in fire-frequented tidal marsh ecosystems. Changes in microbial communities likely relate to changes in microbial ecosystem functions that may be important for the post-fire responses of plants and the larger managed ecosystem. These microbial functions may directly influence plant communities and thereby influence Big Branch Marsh National Wildlife Refuge goals to manage habitat for wading birds and migratory waterfowl. Our study highlights unique, complex microbial responses that are intertwined with fire regimes, cyclone effects and litter loads, sediments and tides, plant communities, substrate heterogeneities, and changes in each effect over time. Understanding responses of communities and their ecosystem functions may be important for management of sensitive marsh ecosystems.

## Supporting information

**S1 File. Diagrams of the fire histories and experimental and plot design.** Fire histories, shown on left, illustrate the date of each fire (red symbol) that occurred over the 10 years prior to the onset of the study in each fire regime (R). The field portion experiment, shown on right, takes place over 6 months from deployment on day 0 to the final collection date in December 2022. Plots and litter bags are classified by fire regime (R), litter load (L), sampling time (D) and collection date. Plots are unpaired and independent despite adjacent placement in table. (DOCX)

**S2 File. Summary table of the plant community composition PERMANOVA analysis.** Permutational multivariate analysis of variance (PERMANOVA) summary table of treatment effects on the plant community composition measured at the onset of the study. Treatment effects were evaluated using Type I sums of squares on interactions and lower terms. Significance:. = $0.05 < p < 0.1$, * = $0.01 < p < 0.05$, ** = $0.001 < p < 0.01$, *** = $p < 0.001$. (DOCX)

**S3 File. Results of all significant contrasts.** Contrasts were performed for significant main and interaction effects for each linear mixed model or Permutational multivariate analysis of variance (PERMANOVA) analysis, and were all compiled and presented here. Under each analysis heading are the significant main or interaction effects identified in the global test for that analysis, that describe the effect, the contrasts, and correction for significance. Under each significant effect are the null hypotheses ($H_0$) for the relevant contrasts, and for each contrast, either a pseudo-F or a chi square statistic, the degrees of freedom for the test, the statistic, and

the p-value. Contrasts were evaluated using Type III sums of squares with Bonferroni corrections applied to significance levels for each set of contrasts evaluated for each analysis. For contrasts involving fire regime, litter load, or their interaction, a significance level of $\alpha = 0.01$ was used; for contrasts involving effects over time, $\alpha = 0.0033$ was used to account for the multiple analyses. Red text indicates a significant contrast and difference.
(DOCX)

**S4 File. Summary table of litter bag mass gain in each treatment combination.** Table of summary statistics of litter bag mass gained per fire regime (R) and litter load (L) treatment at each sampled time point (D). Mean, 95% confidence interval, and lowest and highest mass gains were calculated across all plots within each treatment combination for the 131 total bags that were successfully retrieved.
(DOCX)

**S5 File. Summary table of the litter bag mass gain ANOVA analysis.** Repeated measures Analysis of Variance (ANOVA) table for treatment effects on the mass gained in litter bags. Treatment effects were evaluated using Type III sums of squares on interactions and lower terms. Significance:. $= 0.05 < p < 0.1$, $^{*} = 0.01 < p < 0.05$, $^{**} = 0.001 < p < 0.01$, $^{***} = p < 0.001$.
(DOCX)

**S6 File. Summary bar plots of average mass gained within combinations of fire regimes and litter loads at different times.** Average mass gain in litter bags (vertical bar), and +/- 95% confidence intervals (vertical lines) were calculated and are shown within each combination of fire regime (R) and litter load (L) treatments, at or over (left and right, respectively) the different sampling times (D). Tukey HSD pairwise comparison letter groupings were provided for the comparisons of fire regime and litter load. Treatments with similar letters above bars are statistically similar.
(DOCX)

**S7 File.** Non-metric, multi-dimensional scaling (NMDS) ordination of fungal community compositions within the separated fire regimes of R1 (A), R4 (B), and R5 (C) at different sampling times. Distances and ordination were based on fungal ESV data in each treatment combination. Fungal community compositions are shown here within each fire regime (R), and at different sampling times (D). Plots (symbols) are coded by sampling time by symbol color and shape. Overlain polygons identify the spread and separation of fungal species compositions in plots of the same sampling time.
(DOCX)

**S8 File.** Non-metric, multi-dimensional scaling (NMDS) ordination of fungal community compositions within different litter loads (A), and within different litter loads at different sampling times (B). Distances and ordination were based on fungal ESV data in each treatment combination. Fungal community compositions are shown here within each litter load (L), and at different sampling times (D). Plots (symbols) are coded in panel A by litter load, using symbol color and shape. In panel B, plots are coded by sampling time by symbol shape, and litter load by symbol color. Overlain polygons identify the spread and separation of fungal species compositions in plots of the same litter load, or in plots of the same litter load and sampling time.
(DOCX)

**S9 File.** Non-metric, multi-dimensional scaling (NMDS) ordination of fungal community compositions within the separated litter loads of L1 (A) and L2 (B) at different sampling times. Distances and ordination were based on fungal ESV data in each treatment combination.

Fungal community compositions are shown here within each litter load (L), and at different sampling times (D). Plots (symbols) are coded by sampling time with symbol color and shape. Overlain polygons identify the spread and separation of fungal species compositions in plots of the same sampling time.
(DOCX)

**S10 File. Non-metric, multi-dimensional scaling (NMDS) ordination of fungal community compositions within different fire regimes and litter loads.** Distances and ordination were based on fungal ESV data in each treatment combination. Fungal community compositions are shown here within each combination of fire regime (R) and litter load (L). Plots (symbols) are coded by fire regime with symbol color, and litter load with symbol shape. Overlain polygons identify the spread and separation of fungal species compositions in plots of the same fire regime and litter load combination.
(DOCX)

**S11 File.** Non-metric, multi-dimensional scaling (NMDS) ordination of fungal community compositions within the separated fire regimes of R1 (A), R4 (B), and R5 (C) at different litter loads. Distances and ordination were based on fungal ESV data in each treatment combination. Fungal community compositions are shown here within each fire regime (R), and at different litter loads (L). Plots (symbols) are coded by litter load with symbol color and shape. Overlaid geometric space shows spread and separation of plots of the same litter load.
(DOCX)

**S12 File.** Non-metric, multi-dimensional scaling (NMDS) ordination of fungal community compositions within the separated fire regimes and litter loads of R1L1 (A), R1L2 (B), R4L1 (C), R4L2 (D), R5L1 (E), and R5L2 (F) at different sampling times. Distances and ordination were based on fungal ESV data in each treatment combination. Fungal community compositions are shown here within each fire regime (R) and litter load (L), and at different sampling times (D). Plots (symbols) are coded by sampling time with symbol color and shape. Overlaid geometric space shows spread and separation of plots of the same sampling time.
(DOCX)

**S13 File.** Non-metric, multi-dimensional scaling (NMDS) ordination of bacterial community compositions within the separated fire regimes of R1 (A), R4 (B), and R5 (C) at different sampling times. Distances and ordination were based on bacterial ESV data in each treatment combination. Bacterial community compositions are shown here within each fire regime (R), and at different sampling times (D). Plots (symbols) are coded by sampling time by symbol color and shape. Overlain polygons identify the spread and separation of bacterial species compositions in plots of the same sampling time.
(DOCX)

**S14 File.** Non-metric, multi-dimensional scaling (NMDS) ordination of bacterial community compositions within different litter loads (A), and within different litter loads at different sampling times (B). Distances and ordination were based on bacterial ESV data in each treatment combination. Bacterial community compositions are shown here within each litter load (L), and at different sampling times (D). Plots (symbols) are coded in panel A by litter load, using symbol color and shape. In panel B, plots are coded by sampling time by symbol shape, and litter load by symbol color. Overlain polygons identify the spread and separation of bacterial species compositions in plots of the same litter load, or in plots of the same litter load and sampling time.
(DOCX)

**S15 File.** Non-metric, multi-dimensional scaling (NMDS) ordination of bacterial community compositions within the separated litter loads of L1 (A) and L2 (B) at different sampling times. Distances and ordination were based on bacterial ESV data in each treatment combination. Bacterial community compositions are shown here within each litter load (L), and at different sampling times (D). Plots (symbols) are coded by sampling time with symbol color and shape. Overlain polygons identify the spread and separation of bacterial species compositions in plots of the same sampling time.
(DOCX)

**S16 File. Non-metric, multi-dimensional scaling (NMDS) ordination of bacterial community compositions within different fire regimes and litter loads.** Distances and ordination were based on bacterial ESV data in each treatment combination. Bacterial community compositions are shown here within each combination of fire regime (R) and litter load (L). Plots (symbols) are coded by fire regime with symbol color, and litter load with symbol shape. Overlain polygons identify the spread and separation of bacterial species compositions in plots of the same fire regime and litter load combination.
(DOCX)

**S17 File.** Non-metric, multi-dimensional scaling (NMDS) ordination of bacterial community compositions within the separated fire regimes of R1 (A), R4 (B), and R5 (C) at different litter loads. Distances and ordination were based on bacterial ESV data in each treatment combination. Bacterial community compositions are shown here within each fire regime (R), and at different litter loads (L). Plots (symbols) are coded by litter load with symbol color and shape. Overlaid geometric space shows spread and separation of plots of the same litter load.
(DOCX)

**S18 File.** Non-metric, multi-dimensional scaling (NMDS) ordination of bacterial community compositions within the separated fire regimes and litter loads of R1L1 (A), R1L2 (B), R4L1 (C), R4L2 (D), R5L1 (E), and R5L2 (F) at different sampling times. Distances and ordination were based on bacterial ESV data in each treatment combination. Bacterial community compositions are shown here within each fire regime (R) and litter load (L), and at different sampling times (D). Plots (symbols) are coded by sampling time with symbol color and shape. Overlaid geometric space shows spread and separation of plots of the same sampling time.
(DOCX)

**S19 File. Summary table of the fungal and bacterial community composition PERMANOVA analyses.** Permutational Multivariate Analysis of Variance (PERMANOVA) summary table of treatment effects and error terms on the fungal and bacterial community compositions based on the repeated measures design of the experiment. Treatment effects were evaluated using Type I sums of squares and a significance level of $\alpha = 0.01$. Red font and bolding indicate significant effects with $p < 0.01$.
(DOCX)

**S20 File. Summary table of the fungal and bacterial alpha diversity ANOVA analyses.** Compiled Analysis of Variance (ANOVA) summary tables for the generalized linear mixed models built to evaluate treatment effects on fungal and bacterial richness, evenness, and Shannon diversity. Treatment effects were evaluated using Type III sums of squares, with a significance level of $\alpha = 0.01$. Red font and bolding indicate $p < 0.01$.
(DOCX)

**S21 File.** Boxplots showing alpha diversity metrics within different fire regimes at different sampling times, for fungal species richness (A), fungal evenness (B), fungal Shannon diversity

(C); and bacterial species richness (D), bacterial evenness (E), and bacterial Shannon diversity (F). Boxplots show the alpha diversity metric in each fire regime (R) and sampling time (D) combination. Colors group similar time points across different fire regimes.
(DOCX)

**S22 File.** Boxplots showing alpha diversity metrics within different fire regimes and litter loads at different sampling times, for fungal species richness (A), fungal evenness (B), fungal Shannon diversity (C); and bacterial species richness (D), bacterial evenness (E), and bacterial Shannon diversity (F). Boxplots show the alpha diversity metric in each fire regime (R), litter load (L), and sampling time (D) combination. Colors group similar time points across different fire regimes.
(DOCX)

**S23 File. Spreadsheet of indicator fungi and bacteria in different fire regimes, litter loads, and in different fire regimes at different times.** Fungal and bacterial families were identified as indicators in specific levels of fire regime (R), litter load (L), and combinations of fire regimes and time points (D). Also provided are phi coefficients of association that indicate the strength of positive or negative association of an indicator family with a certain treatment. Higher magnitude indicates stronger association, sign indicates direction. Color indicates strength and direction of association, with red indicating more strongly negative, and green indicating more strongly positive.
(XLSX)

**S24 File. Spreadsheet of summarized results.** Compilation of all analyses on fungal and bacterial communities. Each hypothesis is listed with the corresponding effect tested in each analysis. For the nonmetric multidimensional scaling (NMDS) analysis, visible separation is indicated with the corresponding figure or supplementary file that shows the community patterns for that particular effect. For the Permutational Multivariate Analysis of Variance (PERMANOVA) analysis, indications are provided if the tested effect, corresponding to its specific hypothesis, was found to be significant in the global PERMANOVA test on fungal or bacterial community compositions. For the alpha diversity metrics of richness, evenness, and Shannon diversity, indications are provided if the tested effect, corresponding to its specific hypothesis, was found to be significant in the Analysis of Variance (ANOVA) test of the linear model on a particular metric. For all PERMANOVA and ANOVA tests, contrasts were performed on significant effects identified in the global test, and indications are provided where the corresponding contrasts were significant. For effects in the global test that were not significant, contrasts were not performed. A significance level of $\alpha = 0.01$ was used for all PERMANOVA and ANOVA tests, and for all contrasts involving fire regime, litter load, or their interaction. A significance level of $\alpha = 0.0033$ was used for all contrasts involving effects over time to account for the multiple analyses. Red and bolded font indicate significant effects in each respective test.
(XLSX)

## Acknowledgments

We acknowledge and thank the managers at the Big Branch Marsh National Wildlife Refuge for permission to access and study the marsh. We thank Jimmy Laurent, Daniel Breaux, and Chris LeRouge for providing fire maps and recent history of the marsh. We thank the LSU Health Shreveport Genomics Core for DNA sequencing services.

## Author Contributions

**Conceptualization:** Viet Q. Dao, William J. Platt.

**Data curation:** Viet Q. Dao.

**Formal analysis:** Viet Q. Dao.

**Investigation:** Viet Q. Dao, William J. Platt.

**Methodology:** Viet Q. Dao, William J. Platt.

**Resources:** Crystal N. Johnson.

**Supervision:** Crystal N. Johnson.

**Validation:** Viet Q. Dao, William J. Platt.

**Visualization:** Viet Q. Dao.

**Writing – original draft:** Viet Q. Dao.

**Writing – review & editing:** Viet Q. Dao, Crystal N. Johnson, William J. Platt.

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
