## [Decision Letter · Decision Letter 0]

19 Jun 2024

PONE-D-24-08804Prescribed fire regimes influence responses of fungal and bacterial communities on new litter substrates in a brackish tidal marshPLOS ONE

Dear Dr. Dao,

Thank you for submitting your manuscript to PLOS ONE. After careful consideration, we feel that it has merit but does not fully meet PLOS ONE’s publication criteria as it currently stands. Therefore, we invite you to submit a revised version of the manuscript that addresses the points raised during the review process. Analyzing the manuscript and the revisions made from the two anonymous reviewers, I believe that the manuscript (ms) has good scientific quality to be published in PLOS ONE. However, the length of the ms turns its readability extremely hard and confusing. I recommend the reduction of size manuscript according to the suggestion of the reviewers. A completely agree that the methods section should be reduced considerable and due to the amount of data and the consequent interpretation, the merge of the results and discussion section will be very helpful.

We look forward to receiving your revised manuscript.

Kind regards,

João Canário, PhD

Academic Editor

PLOS ONE

Journal Requirements:

4. We note that Figure S1 in your submission contain [map/satellite] images which may be copyrighted. All PLOS content is published under the Creative Commons Attribution License (CC BY 4.0), which means that the manuscript, images, and Supporting Information files will be freely available online, and any third party is permitted to access, download, copy, distribute, and use these materials in any way, even commercially, with proper attribution. For these reasons, we cannot publish previously copyrighted maps or satellite images created using proprietary data, such as Google software (Google Maps, Street View, and Earth). For more information, see our copyright guidelines: http://journals.plos.org/plosone/s/licenses-and-copyright.

a. You may seek permission from the original copyright holder of Figure S1 to publish the content specifically under the CC BY 4.0 license.  

Reviewers' comments:

Reviewer's Responses to Questions

**Comments to the Author**

1. Is the manuscript technically sound, and do the data support the conclusions?

Reviewer #1: Yes

Reviewer #2: Yes

2. Has the statistical analysis been performed appropriately and rigorously? 

Reviewer #1: Yes

Reviewer #2: Yes

3. Have the authors made all data underlying the findings in their manuscript fully available?

Reviewer #1: Yes

Reviewer #2: Yes

4. Is the manuscript presented in an intelligible fashion and written in standard English?

Reviewer #1: Yes

Reviewer #2: Yes

5. Review Comments to the Author

Reviewer #1: This is a very interesting study that deals with the changes in litter subtract dynamics, fire regimes and impact on microbial and fungi communities.

To the best of my knowledge, this is the first time those variables are sutided in coastal environmental.

The manuscript is scientifically correct and discussion and conclusions are inline with the obtained results.

My main concerns are not related with the science produced by the authors but with the readability of the manuscript.

I had to read it seven times for making this revision because the amount of data, and methodologies are so extended that the reader will lose very easily the fluidity of manuscript.

I believe that before this manuscript should be considered for publication, a profound revision should be done in order to reduce the length of the manuscript. !! pages of methods is undoublty to large. The number of figures and the length of the results section is enormous and this turns the discussion very confusing. I will suggest merge the results and discussion sections.

I will be happy to review the revised version of this manuscript.

The quality of the science deserves this revision.

Reviewer #2: The manuscript entitled "Prescribed fire regimes influence responses of fungal and bacterial communities on new litter substrates in a brackish tidal marsh" by Dao et al. presented the effects of litter types in different fire regimes and litter loads on microbial community composition and changes over time in the context of frequent tidal inundation in a coastal brackish marsh.

This study investigates the impact of different fire regimes on microbial communities, specifically focusing on fungal and bacterial families, within coastal marsh ecosystems, and could be summarized in several key points:

1) Fire Regimes and Microbial Communities: Different fire regimes (e.g., frequency and intensity of fires) had substantial and similar impacts on both fungal and bacterial indicator families and their community compositions. The changes in microbial communities followed distinct developmental trajectories over time depending on the fire regime.

2) Disconnection from Plant Communities: Despite the influence of fire regimes on microbial communities, these changes were not directly correlated with variations in plant communities, indicating a decoupling of plant and microbial community responses to fire.

3) Role of Litter Loads and Tidal Inundation: Variations in litter loads due to different fire regimes led to differences in sediment accumulation, which is influenced by tidal inundation patterns. These sediment differences potentially affect microbial community structures.

4) Interaction with Tropical Cyclones: The study suggests an interactive effect between fire regimes and tropical cyclones, both of which contribute to substrate heterogeneities. Such substrate heterogeneities can alter the composition of microbial communities, which in turn may affect the accumulation of fine fuels and influence subsequent fire events.

5) Implications for Ecosystem Management: Understanding how microbial communities respond to fire regimes and tropical cyclones is crucial for the management of coastal marsh ecosystems. These insights can help in predicting and mitigating the impacts of fire and extreme weather events on these ecosystems.

In summary, the study highlights the significant influence of fire regimes on microbial communities and underscores the complexity of interactions between fire, cyclones, and tidal processes in shaping coastal marsh ecosystems. These findings are important for developing effective management strategies aimed at preserving the ecological balance and resilience of these environments.

However, the manuscript presented a vast study where several number of variables were studied at the same time, leading to an extremely big document to read. Therefore, readers may feel a little lost when faced with so many variables to be considered and discussed simultaneously. Sometimes it becomes difficult to read from beginning to end without getting lost in the middle. Maybe the authors should considered to reduce the manuscript (see comment bellow) in order to present a more easier reading and comprehensive one.

In general, the manuscript is very well written, and the sections are very complete according to the proposed. Material and Methods section is extensively well explained with all the sub-sections indicated very detailed. I suggest to reduce a bit this section in order to reduce the size of the manuscript. Too much detail is not necessary in the main manuscript and could be in a supplementary support file.

6. PLOS authors have the option to publish the peer review history of their article (what does this mean?). If published, this will include your full peer review and any attached files.

Reviewer #1: No

Reviewer #2: No

---

## [Author Response · Author response to Decision Letter 0]

16 Aug 2024

All responses are provided in the Response To Reviewers! Overall, the manuscript has undergone drastic revision and restructuring. The bulk of edits focused on removing accessory information that was ultimately unnecessary for the main narrative of the manuscript, and focusing discussion and mention onto information that was biologically significant. Other problematic parts such as data not shown, or copyright issues, have been addressed as well.

Thank you all for your edits and comments. We welcome more revisions if you have any!

---

## [Decision Letter · Decision Letter 1]

17 Sep 2024

Prescribed fire regimes influence responses of fungal and bacterial communities on new litter substrates in a brackish tidal marsh

PONE-D-24-08804R1

Dear Dr. Dao,

We’re pleased to inform you that your manuscript has been judged scientifically suitable for publication and will be formally accepted for publication once it meets all outstanding technical requirements.

Kind regards,

João Canário, PhD

Academic Editor

PLOS ONE

Additional Editor Comments (optional):

The manuscript was greatly improved with the revision. The authors made it smaller and focuses on the essential concepts, results and discussion. I still believe that the merge of the results and discussion session would make the work more readable but I accept the comments of the authors concerning this subject and the ok from both reviewers. 

Reviewers' comments:

Reviewer's Responses to Questions

**Comments to the Author**

1. If the authors have adequately addressed your comments raised in a previous round of review and you feel that this manuscript is now acceptable for publication, you may indicate that here to bypass the “Comments to the Author” section, enter your conflict of interest statement in the “Confidential to Editor” section, and submit your "Accept" recommendation.

Reviewer #1: All comments have been addressed

Reviewer #2: All comments have been addressed

2. Is the manuscript technically sound, and do the data support the conclusions?

Reviewer #1: Yes

Reviewer #2: Yes

3. Has the statistical analysis been performed appropriately and rigorously? 

Reviewer #1: Yes

Reviewer #2: Yes

4. Have the authors made all data underlying the findings in their manuscript fully available?

Reviewer #1: Yes

Reviewer #2: Yes

5. Is the manuscript presented in an intelligible fashion and written in standard English?

Reviewer #1: Yes

Reviewer #2: Yes

6. Review Comments to the Author

Reviewer #1: The authors made a strong effort to reduce the size of the manuscript and make it more readable. The revision version has much better quality that the initial one. I still feel that merging the results and discussion section would improve the fluidity of the paper but I accept the arguments from the authors.

Reviewer #2: The manuscript was successfuly updated and the improvements were very well done. Nevertheless, the authors created a support file with all the information that could be useful to readers but that does not necessarily need to be in the main text, which was a good choice. The new improved version of the manuscript presented a main text which has been reduced considerably and all suggestions, questions and doubts raised by the reviewers have been considered and responded accordingly.

7. PLOS authors have the option to publish the peer review history of their article (what does this mean?). If published, this will include your full peer review and any attached files.

Reviewer #1: No

Reviewer #2: No

---

## [Editor Report · Acceptance letter]

23 Sep 2024

PONE-D-24-08804R1 

PLOS ONE

Dear Dr. Dao, 

I'm pleased to inform you that your manuscript has been deemed suitable for publication in PLOS ONE. Congratulations! Your manuscript is now being handed over to our production team.

Kind regards, 

on behalf of

Dr. João Canário 

Academic Editor

PLOS ONE